# Universal scaling law for chiral antiferromagnetism

Shijie Xu[1,2,3,4,5,7], Bingqian Dai[2,7], Yuhao Jiang[1,3,7], Danrong Xiong[1,4], Houyi Cheng[1,4,5], Lixuan Tai[2,7], Meng Tang[3], Yadong Sun[3], Yu He[1,4], Baolin Yang[6], Yong Peng[6], Kang L. Wang[2] & Weisheng Zhao[1,4,5] ✉

The chiral antiferromagnetic (AFM) materials, which have been widely investigated due to their rich physics, such as non-zero Berry phase and topology, provide a platform for the development of antiferromagnetic spintronics. Here, we find two distinctive anomalous Hall effect (AHE) contributions in the chiral AFM $Mn_3Pt$, originating from a time-reversal symmetry breaking induced intrinsic mechanism and a skew scattering induced topological AHE due to an out-of-plane spin canting with respect to the Kagome plane. We propose a universal AHE scaling law to explain the AHE resistivity ($\rho_{AH}$) in this chiral magnet, with both a scalar spin chirality (SSC)-induced skew scattering topological AHE term, $a_{sk}$ and non-collinear spin-texture induced intrinsic anomalous Hall term, $b_{in}$. We found that $a_{sk}$ and $b_{in}$ can be effectively modulated by the interfacial electron scattering, exhibiting a linear relation with the inverse film thickness. Moreover, the scaling law can explain the anomalous Hall effect in various chiral magnets and has far-reaching implications for chiral-based spintronics devices.

Chirality and chirality-induced novel phenomena are commonly observed and extensively studied in chiral spintronics[1,2]. In particular, an ultra-high tunneling magnetoresistance (TMR) of 300% and a spin polarization of 60% are obtained in chiral polymer materials due to chirality-dependent tunneling current[3]. In addition, a chiral anomaly in a Weyl band structure populates Weyl fermions in $Mn_3Sn$ with a specific chirality, which gives rise to a chiral current and a characteristic negative magnetoresistance[4,5]. The Kagome lattice of chiral AFM $Mn_3X$ (such as $Mn_3Sn$[6], $Mn_3Pt$[7], $Mn_3Ge$[8], $MnGe$[9], $Mn_3Ir$[10]) also displays a large AHE that arises from the non-zero berry phase from Bloch bands in momentum space. The AHE in high-quality $Mn_3Sn$ and $Mn_3Ge$ single crystals originates from the intrinsic mechanism[11]. However, the giant AHE in chiral AFM $MnGe$ exceeds the conventional quantum limits of intrinsic AHE due to the skew scattering

AHE[9,12,] which has a dominant contribution. Moreover, $Mn_3Sn$ and $MnGe$ exhibit different scaling relationships[9,11], with their anomalous Hall conductance linearly depending on or independent of the conductance value, respectively.

The different scaling laws in $Mn_3Sn$ and $MnGe$ originate from two distinct physical mechanisms. The two-spin correlation[13,14] is a good method to explain the large intrinsic AHE in $Mn_3Sn$, by the vector spin chirality[15,16](VSC) $\varepsilon = S_1 S_2 + S_2 S_3 + S_3 S_1$. For example, non-collinear (collinear) $Mn_3Pt$ exhibit non-zero (zero) VSC, respectively, leading to non-zero (zero) intrinsic anomalous Hall response[7]. In addition, $Mn_3Sn$, $Mn_3Ir$, and $Mn_3Ge$ ($Mn_3Pt$, $Mn_3Ga$) exhibit negative (positive) VSC[1,2], leading to negative (positive) intrinsic anomalous Hall conductance[6–8,11,17,18]. Besides intrinsic AHE for chiral AFM, skew scattering topological AHE can be induced by the three-spin

[1]National Key Laboratory of Spintronics, Hangzhou International Innovation Institute, Beihang University, 311115 Hangzhou, China. [2]Department of Electrical and Computer Engineering, University of California, Los Angeles, CA 90095, USA. [3]Shanghai Key Laboratory of Special Artificial Microstructure, Pohl Institute of Solid State Physics and School of Physics Science and Engineering, Tongji University, Shanghai 200092, China. [4]Fert Beijing Institute, School of Integrated Circuit Science and Engineering, Beihang University, Beijing 100191, China. [5]Hefei Innovation Research Institute, Beihang University, Hefei, China. [6]School of Materials and Energy, or Electron Microscopy Centre of Lanzhou University, Lanzhou University, Lanzhou 730000, P. R. China. [7]These authors contributed equally: Shijie Xu, Bingqian Dai, Yuhao Jiang, Lixuan Tai. ✉e-mail: weisheng.zhao@buaa.edu.cn

correlated scalar spin chirality[16,19] (SSC):$(S_1 \times S_2) \cdot S_3$. The skew scattering topological AHE exceeds the threshold value of the quantization limit [9,20,21] ($-\frac{e^2}{ha}$ for MnGe). Here, $a = 3.83$ Å is the lattice constant for MnGe. Therefore, the intrinsic AHE in MnGe is negligible relative to the skew scattering topological AHE, resulting in an entirely different scaling relationship from $Mn_3Sn$.

Here, we systematically study the AHE of the chiral non-collinear AFM $Mn_3Pt$ and find a universal AHE scaling law. This anomalous Hall resistivity $\rho_{AH}$ can be described by[20,22]

$$\rho_{AH} = a_{sk}\rho_{xx} + b_{in}\rho_{xx}^2 \qquad (1)$$

where $\rho_{xx}$ is the longitudinal resistivity. The first term $a_{sk}\rho_{xx}$ is the scalar spin chirality-induced skew scattering topological AHE[9,12]. The second term $b_{in}\rho_{xx}^2$ describes the intrinsic anomalous Hall effect arising from time symmetry breaking by the compensated non-collinear magnetic order[8,17,18]. By carefully designing an experimental procedure for $Mn_3Pt$ alloy films, we find that the AHE parameter $a_{sk}$, $b_{in}$ and sheet resistively $\rho_{xx}$ change linearly with the inverse film thickness $d$. The linear 1/d dependencies could be attributed to the symmetry breaking at the surface. In addition, the scaling law can explain the anomalous Hall effect in various chiral magnets ($Mn_3Sn$[6], $Mn_3Pt$[7], $Mn_3Ge$[8], MnGe[9], $Mn_3Ir$[10]) and should be universal for describing the AHE of chiral magnets.

## Results and discussions
### Chiral magnet $Mn_3Pt$ with single crystal properties and topological spin texture

$Mn_3Pt$ is a cubic chiral antiferromagnetic intermetallic compound with lattice constant $a = 3.833$ Å[23]. It exhibits chiral AFM spin order below the Néel temperature $T_N \approx 474$ K[23–25]. We epitaxially grow high-quality $Mn_3Pt$ thin film by sputtering on MgO (001) single-crystal substrate. Typical X-ray diffraction (XRD) spectra of MgO substrate and $Mn_3Pt$ are measured at room temperature. The out-of-plane XRD theta to 2theta scans at different planes indicate that $Mn_3Pt$ film on the MgO substrate is a single crystal (Fig. 1a). The two extra peaks of $Mn_3Pt$ (111) and (222), which indicates the establishment of the long-range chemical ordering, are located at $2\theta_1 = 40.5°$ and $2\theta_1 = 87.5°$. 360° phi scan around $Mn_3Pt$ (111) plane and MgO (111) was measured by XRD to ensure the single-crystal property (Fig. 1b). The MgO and the $Mn_3Pt$ show fourth-degree symmetry, indicating the epitaxial growth of the sample. As a non-collinear AFM, $Mn_3Pt$ has a weak magnetic moment at room temperature due to symmetry-allowed spin canting. In addition, the spin canting induces a net moment at 0 T as shown by the measurement of the out-of-plane magnetic hysteresis loop (Fig. 1c). The direction of spin canting is not parallel to the Kagome lattice plane, which is similar to the previous report[7]. The corresponding spin texture at different fields is shown in the illustration figure (Fig. 1c). Here, the SSC is characterized by the stacking direction of the atoms as viewed from [111] axis. When the external magnetic field is positive (greater than the saturation field), the antiferromagnetic spin structure will tilt upward. As a sequence, the SSC will be positive and have the opposite result under negative magnetic field. In addition, when the magnetic field is zero, the net moment and SSC will be non-zero due to magnetic hysteresis effect which will induce a topological AHE.

The scanning transmission electron microscopy (STEM studies) further confirms the high-quality growth of chiral magnet $Mn_3Pt$. The interlayer spacing is 3.8 Å in $Mn_3Pt$ film (Fig. 1d), which is consistent with the XRD studies. In addition, high-resolution STEM and the corresponding selected-area electron diffraction (SAED) studies reveal the single crystal feature of the chiral magnet $Mn_3Pt$ (Fig. 1e, f), which is essential for achieving the designed properties and functional spintronics devices[26–29].

## Bulk-like anomalous Hall effect and temperature dependence of transport properties

To better understand the topological spin chirality induced zero field Hall response. We prepare chiral magnet $Mn_3Pt$ films on MgO substrate with different thickness values by dc magnetron sputtering and measure the resistance by the standard four-point measurement set-up (Fig. 2a). At 340 K, the Anomalous Hall effect in the $Mn_3Pt$ films changes with different thickness d (Fig. 2b). The AHE coercive field decreases with the film thicknesses $d$, which is consistent with the reduction of the coercive field in the $Mn_3Pt$ (t nm)/STO substrate[7]. It is worth noting that the magnetic coercivity is much smaller than that of AHE coercivity, because the Zero magnetic moment indicates that the spin canting is zero. However, the AHE coercivity indicates that the topological AHE and the intrinsic AHE have opposite and equal contributions. The anomalous Hall resistivity ($\rho_{AH} = \frac{(\rho_{xy}^+ - \rho_{yx}^-)}{2}$) is 0.086, 0.114, 0.156, 0.346, 0.465 μΩ*cm, when the thickness is 12, 14, 18, 22, 30 nm, respectively, showing the bulk like anomalous Hall effect.

Temperature-dependent studies of 30 nm $Mn_3Pt$ show that the anomalous Hall resistivity $\rho_{AH}$ becomes smaller at higher temperature and change monotonically with the temperature (Fig. 2c). The longitudinal resistivity $\rho_{xx}$ gradually increases at the higher temperature, indicating the metallic behavior (Fig. 2c), and there is no phase transition during our measurement. In addition, the $\rho_{xx}$ (T) can be fitted by a linear function of $T^2$ due to the dominant electron scattering by the spin flip[30,31] mechanism. The $\rho_{AH}$ is about three orders of magnitude smaller than the resistivity $\rho_{xx}$, and we can get the temperature dependence of AHE angle by the equation $\theta_{AH} = \frac{\rho_{AH}}{\rho_{xx}}$ (Fig. 2d). The anomalous Hall angle is −0.168% at 300 K, and the magnetic moment due to spin canting still have the contribution for SSC excitation at zero field.

## Universal AFM scaling law for chiral AFM $Mn_3Pt$

Magnetic materials are the best choice to explore the AHE scaling law. Previous studies have achieved great success by attributing the AHE in ferromagnetic materials to both intrinsic and extrinsic contributions[20,22]. To extract the different contributions of the AHE in chiral AFM magnets with different thicknesses $d$, we use the AFM scaling law $\rho_{AH} = a_{sk}\rho_{xx} + b_{in}\rho_{xx}^2$, to describe the measured data. $\rho_{AH}$ is plotted as a function of $\rho_{xx}$ (Fig. 3, illustration picture), the variations of $\rho_{AH}$ and $\rho_{xx}$ are achieved by varying temperatures for a given sample. Clearly, $a_{sk}$ arises from the linear $\rho_{xx}$ dependence and $b_{in}$ arises from the quadratic $\rho_{xx}$ dependence (illustration picture). The quadratic term represents the intrinsic Hall effect, while the linear term represents the skew scattering mechanism determined by the scalar spin chirality. To extract different contributions to the AHE, the $\frac{\rho_{AH}}{\rho_{xx}}$ ratio is plotted as a function of $\rho_{xx}$ (Fig. 3). In this case, the slope and intercept represent the intrinsic AHE parameter $b_{in}$ and skew scattering AHE parameter $a_{sk}$, respectively. The positive skew scattering AHE of $Mn_3Pt$ has an opposite sign to the intrinsic Hall effect, which has been predicted by recently Boltzmann transport theory in chiral magnet[12]. To ascertain the universal AFM scaling law for chiral AFM, we plot the AHE data for various non-collinear AFM $Mn_3X$ (X = Sn[6], Pt, Ge[8], Ir[10], etc.) and MnGe[9]. For chiral AFM MnGe, the anomalous Hall angle ($\theta_{AH} = \frac{\rho_{AH}}{\rho_{xx}}$) is −0.18, much larger than other chiral AFM. In addition, the intrinsic Hall resistivity term ($b_{in}\rho_{xx}^2$) of MnGe is negligible relative to the skew scattering topological anomalous Hall resistivity term ($a_{sk}\rho_{xx}$). In this situation, the scaling law can be simplified to $\rho_{AH} = a_{sk}\rho_{xx}$ and $\sigma_{AH} = a_{sk}\sigma_{xx}$ [$\sigma_{xx} = \frac{\rho_{xx}}{\rho_{xx}^2 + \rho_{xy}^2}$, $\sigma_{AH} = \frac{-\rho_{xy}}{\rho_{xx}^2 + \rho_{xy}^2}$]. Therefore, the anomalous Hall conductance $\sigma_{AH}$ will have a linear relationship with longitudinal conductance $\sigma_{xx}$ [9]. On the contrary, for chiral magnets $Mn_3Sn$ and $Mn_3Ge$ without out-plane spin canting, recent work[11] shows that the skew scattering AHE resistivity term ($a_{sk}\rho_{xx}$) is negligible relative to intrinsic anomalous Hall resistivity term ($b_{in}\rho_{xx}^2$), the scaling

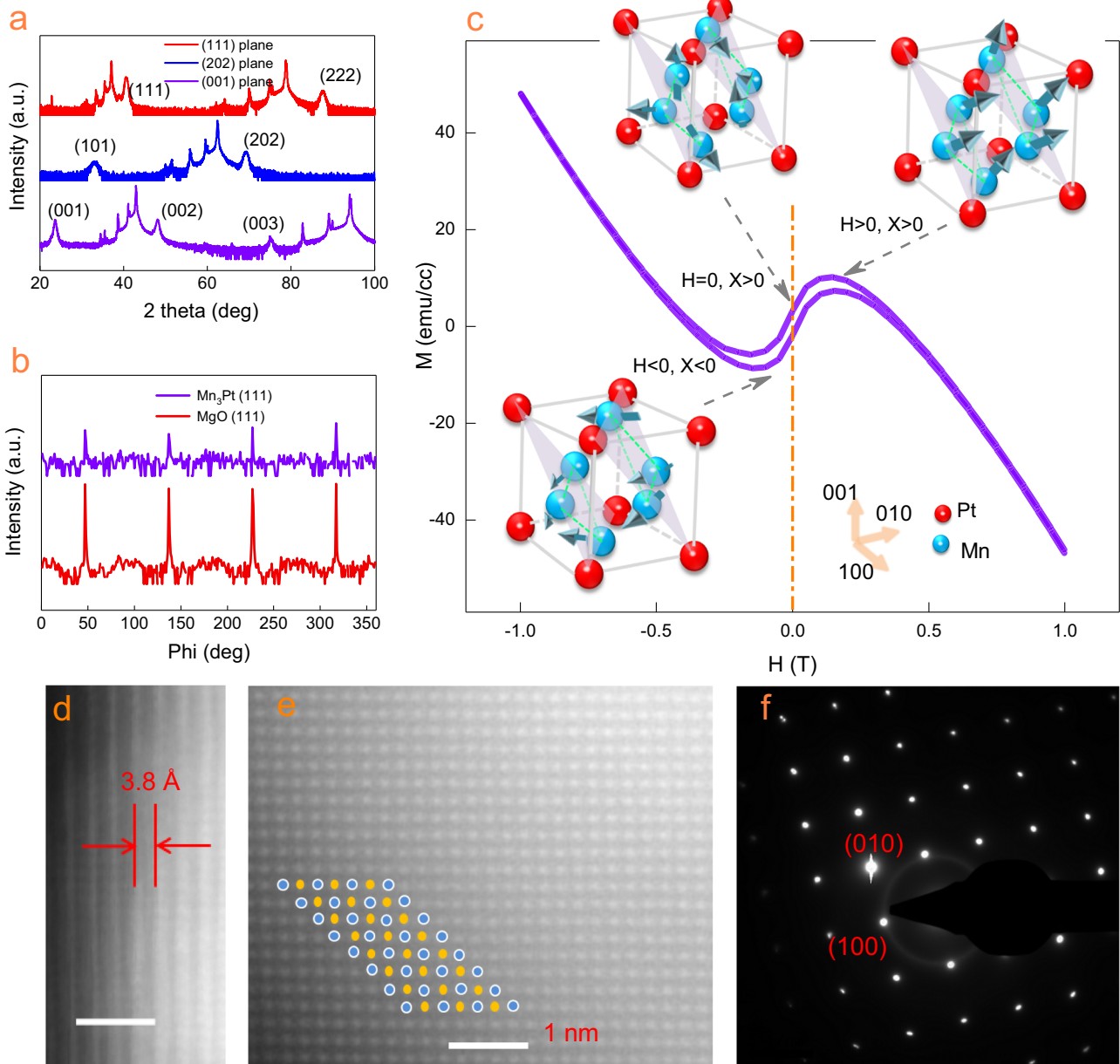

**Fig. 1 | structure of Mn₃Pt film. a** Out-of-plane theta to 2theta XRD spectra of Mn₃Pt at different planes. **b** Phi scan patterns of Mn₃Pt at (111) plane and the corresponding MgO substrate. **c** The magnetization data was measured at 300 K with the magnetic field paralleled to the c-axis. The spin structure of cubic Mn₃Pt is shown in the inset (the red sphere denotes Pt atom, the blue sphere and arrow denote Mn atom and spin direction, respectively. Mn atoms at the Kagome planes have different scalar spin chirality under different magnetic fields.

**d** High-resolution cross-sectional scanning transmission electron microscopy (STEM) images of Mn₃Pt/MgO. The bright regions correspond to the Mn₃Pt monolayers. The scale bar is 1 nm. **e** High-resolution STEM image of the basal plane of the Mn₃Pt. Blue and yellow dots represent the Pt atom and the Mn atom, respectively. The scale bar is 1 nm. **f** Selected-area electron diffraction (SAED) patterns of Mn₃Pt obtained by transmission electron microscopy (TEM).

law can be simplified to $\sigma_{AH} = \frac{-\rho_{xy}}{\rho_{xx}^2} = b_{in}$. Therefore, the intrinsic anomalous Hall conductance $\sigma_{AH}$ will be a constant value for Mn₃Sn[11] and the slope term $a_{sk}$ due to SSC-induced skew scattering is zero (Fig. 3). However, for the chiral AFM Mn₃Ir and Mn₃Pt, both the VSC-induced intrinsic AHE and SSC-induced skew scattering AHE have strong contributions, but the AHE of all the chiral AFM can be well explained by this AFM scaling law.

## Theoretical understanding

In most studies of the AHE scaling law, the interfacial and bulk resistivity are often equally treated[9,11]. However, when the film's thickness is comparable to an electron's mean free path, the anomalous Hall

resistivity and the sheet resistivity will change with the film thickness[22]. $a_{sk}$, $b_{in}$, measured magnetizations M and sheet resistivity $\rho_{xx}$ could be described as a function of the thickness (Fig. 4a–c) which change linearly with 1/d, where the intercept and the slope correspond to bulk and surface contribution, respectively. The magnetizations M induced by the net moment increases with the thickness, which means that the spin structure will tilt larger with the thickness.

As a result, the SSC will increase at larger thickness, causing the larger topological anomalous Hall effect. The intrinsic AHE parameter $b_{in}$ tripled when thickness increased from 12 to 30 nm, indicating that the interfacial electron scattering can effectively control the intrinsic scattering and the presence of stronger interfacial scattering will reduce the AHE in chiral AFM. It is worth noting that this tuning effect

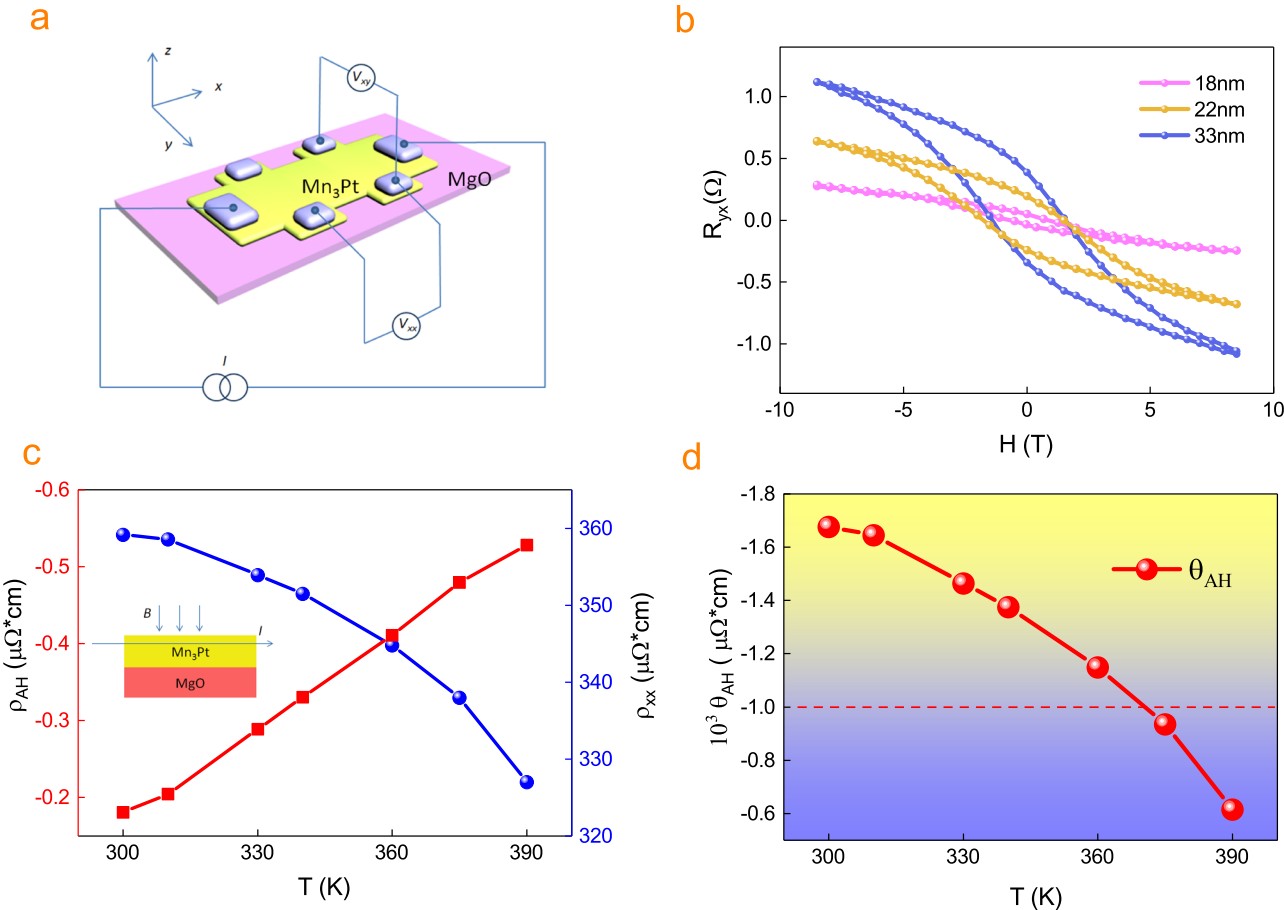

**Fig. 2 | Magnetic field dependence of the AHE in Mn₃Pt. a** 3D schematic of the Hall bar structure made from the Mn₃Pt (yellow)/MgO (purple) stack with a top gate electrode (light gray). Standard four-point measurement setup is displayed. **b** Magnetic-field-dependent anomalous Hall resistance at different thicknesses for chiral magnets Mn₃Pt, the measured temperature is 340 K. **c** Temperature dependence of $\rho_{xx}$ and $\rho_{xy}$ for 30 nm Mn₃Pt. **d** The anomalous Hall angle $\theta_{AH} = \frac{\rho_{AH}}{\rho_{xx}}$ for 30 nm Mn₃Pt.

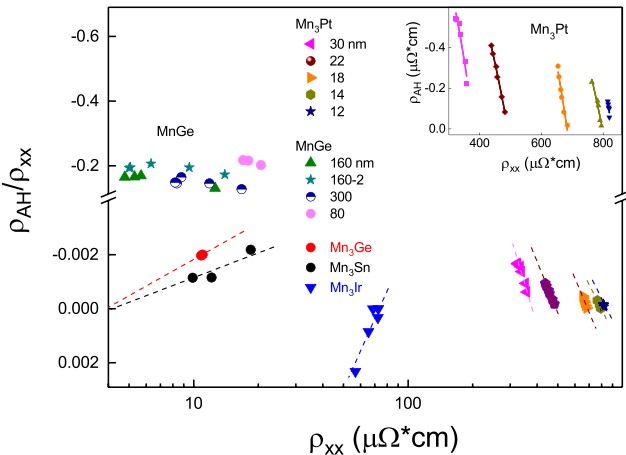

**Fig. 3 | Universal scaling law for chiral magnets.** The various chiral AFM was plotted by $\frac{\rho_{AH}}{\rho_{xx}} = a_{sk} + b_{in}\rho_{xx}$. The slope and intercept represent the intrinsic and skew scattering anomalous Hall scaling factors, respectively. The data includes Mn₃Ge[11], Mn₃Sn[11], Mn₃Ir[10], MnGe[9], and Mn₃Pt. For Mn₃Pt (12 ~ 30 nm) /MgO substrate, the inset is the measured anomalous Hall resistance $\rho_{AH}$ as a function of resistivity $\rho_{xx}$. Solid lines refer to the scaling law fitting of $\rho_{AH} = a_{sk}\rho_{xx} + b_{in}\rho_{xx}^2$.

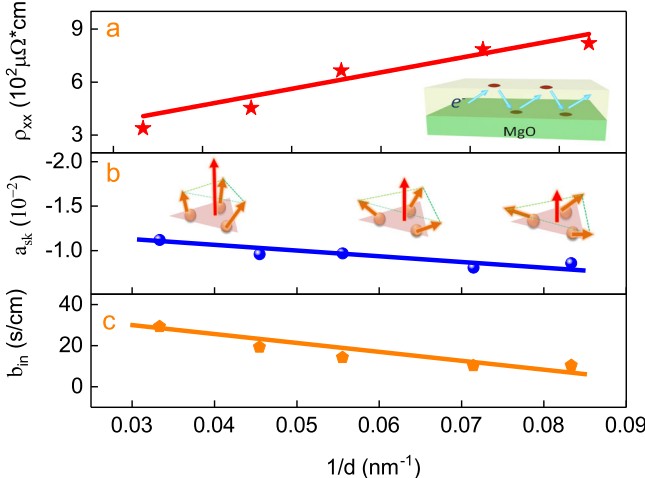

**Fig. 4 | Thickness dependence at room temperature. a–c** The sheet resistivity $\rho_{xx}$, $a_{sk}$, the measured magnetizations M and $b_{in}$ versus 1/d for Mn₃Pt. The inset in (**a**) shows the electronic scattering at the surface and the interface. The inset of (**b**) illustrates that there is larger SSC excitation in thicker films. The measurements were performed at 340 K. Solid lines refer to the linear fitting results. Five samples were measured for each thickness and each error bar indicates the standard error of the mean for the five samples of each thickness.

is particularly weak when the AFM thickness is much larger than the mean free path. In addition, the anomalous Hall angle $\theta_{AH}$ achieves larger values (Fig. 4a–c) for thicker samples with larger values of $a_{sk}$ and $b_{in}$.

The thickness-controlled AHE can also be understood by the following mechanism. The larger out-of-plane anisotropy due to larger spin canting in thicker film (Fig. 2b) allows a larger SSC to be excited, as schematically illustrated in the insets of Fig. 4b. Hence, we can achieve a larger skew scattering topological AHE. To corroborate our result, we also perform first principle studies by calculating the relationship between the intrinsic anomalous Hall conductance (AHC) and SSC induced by out-of-plane spin canting. To illustrate the effect of SSC, we fix the lattice constant to be 3.83Å to exclude the influence of strain. Then, we perform the constrained magnetization calculations. The magnetic moment is initially set to a purely antiferromagnetic state, then we slightly tilt the magnetization of Mn atoms, the total net magnetization is set to the [111] direction of the $Mn_3Pt$ lattice, as shown in (Fig. 1c). We use the component of the total magnetization in the z-direction to represent the state, and the z-component varies from $0.00\mu_B$ to $0.30\mu_B$. The band structure does not change much in most spaces in the Brillouin zone (BZ). However, when a net magnetization exists, the bands at $\Gamma(0,0,0)$-R(0.5,0.50.5) path get closer and thus lead to a higher intrinsic AHE. We plot the spin polarization along the

[111] direction at different k points in (Fig. 5b). The spin direction at k-points on the Γ-R path near the Fermi level have clear tendency to [111] direction. [111]-oriented net magnetization will shift the bands and gives rise to a dramatic increase of the Berry curvature at the k-point between these two bands (Fig. 5a, b). Thus, a weak net magnetization could enhance the intrinsic AHC of $Mn_3Pt$. The calculated intrinsic contribution to the AHC versus Fermi energy with different net magnetizations is shown in (Fig. 5c). The calculation results show that the intrinsic AHC $\sigma_{in}$ is increased from 56.73 $\Omega^{-1}cm^{-1}$ to 73.90 $\Omega^{-1}cm^{-1}$ when the z−component of net magnetization increases from 0 to 0.30 $\mu_B$ (Fig. 5e, f). This calculation elucidates our result and the origin of the chirality-induced anomalous Hall effect, which has been recently reported in chiral magnets[1–13].

## Conclusions

The AHE in chiral antiferromagnet is mainly derived from the three-spin model correlated with SSC[15,16] and the two-spin model correlated with VSC[13,14]. In our experiment, we simultaneously discover the vector spin chirality-induced intrinsic anomalous Hall effect and the scalar spin chirality skew scattering topological AHE in chiral AFM $Mn_3Pt$ and provide a universal AHE scaling law to explain both $Mn_3Pt$ and other chiral AFMs. This work also elucidates the origin of the chirality-induced anomalous Hall effect by ab initio calculations. Our work

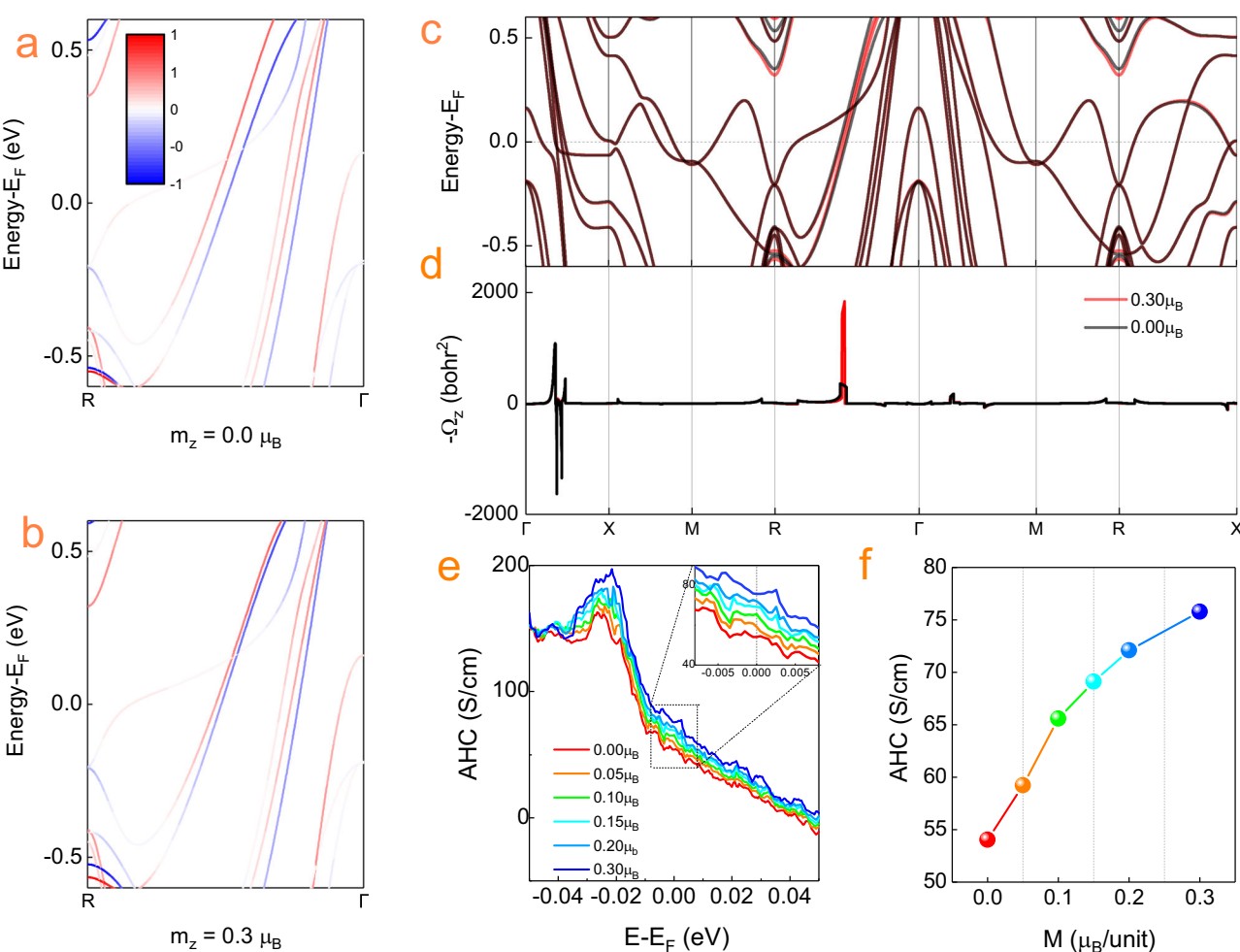

**Fig. 5 | Theoretical calculations of Berry curvature and AHE. a, b** Band structure of R–Γ path for $Mn_3Pt$, the color represents the spin polarization component on [111] direction. **c, d** Band structure near Fermi energy (upper panel) and Berry curvature (lower panel) in atomic units along the symmetry lines. The black line represents the purely antiferromagnetic state, the red line represents the state with net magnetization along [111] direction, whose z-component is $0.30\mu_B$. **e** Calculated intrinsic contribution to the AHC versus Fermi energy with different net magnetizations. **f** Calculated intrinsic contribution to the AHC versus different total magnetizations at the Fermi level.

deepens the understanding of the anomalous Hall effect in antiferromagnetic materials and facilitates the development of chiral spintronic devices.

## Methods

### Material growth

$Mn_3Pt$ thin films were sputtered from a $Mn_3Pt$ target onto (001)-oriented MgO single-crystal substrates ($10 \times 10 \times 0.5$ mm$^3$) with a base pressure of $5 \times 10^{-6}$ Pa. Te deposition was performed at 600 °C. The sputtering power and Ar gas pressure were 30 W and 0.5 Pa, respectively. The deposition rate was $1 \, \text{Å s}^{-1}$, as determined by X-ray reflectivity measurements. After deposition, $Mn_3Pt$ films were kept at 600 °C in a vacuum for annealing for 1 h.

### XRD

XRD measurements were performed by a Bruker D8 diffractometer with a five-axis configuration and Cu Kα ($\lambda = 0.15419$ nm).

### STEM

Cross-sectional wedged samples were prepared by mechanical thinning, precision polishing and ion milling. An electron-beam probe was utilized to scan thin films to achieve high resolution of local regions.

### Electrical measurements

Electrical contacts onto the $Mn_3Pt$ films were made by Al wires via wire bonding. Electrical measurements were performed in a Quantum Design physical property measurement system. The electrical current used for both longitudinal and Hall resistance measurements was 500 μA.

### Magnetic measurements

Magnetic measurements were performed in a Quantum Design superconducting quantum interference device magnetometer with $10^{-11}$ A.m$^{-2}$ sensitivity.

### First-principles calculations

The ab initio calculations were performed using the QUANTUM ESPRESSO package based on the projector augmented-wave method and a plane-wave basis set[32,33]. The exchange and correlation terms were described using a generalized gradient approximation in the scheme of Perdew-Burke-Ernzerhof parametrization, as implemented in the PSLIBRARY[34]. The fully relativistic pseudopotential was used, and the spin-orbit coupling was included in our calculation. A k-point mesh of $20 \times 20 \times 20$ was used in the self-consistent calculations. Then the plane-wave functions were transferred to the maximally localized Wannier functions using the WANNIER90 package[35]. The intrinsic contribution of AHC was calculated by integrating the Berry curvature over the occupied bands through the whole Brillouin zone (BZ)[36].

$$\sigma_{xy} = -\frac{e^2}{\hbar} \int_{BZ} \frac{d\boldsymbol{k}^3}{(2\pi)^3} \sum_n f_n \Omega_n^z(\boldsymbol{k}) \tag{2}$$

and the Berry curvature could be given in a Kubo formula term

$$\Omega_n^z(\boldsymbol{k}) = \hbar^2 \sum_{m \neq n} \frac{-2Im\langle \psi_{n\boldsymbol{k}} | v_x | \psi_{m\boldsymbol{k}} \rangle \langle \psi_{m\boldsymbol{k}} | v_y | \psi_{n\boldsymbol{k}} \rangle}{(E_{n\boldsymbol{k}} - E_{m\boldsymbol{k}})^2} \tag{3}$$

where $v$ are velocity operators. An ultra-dense k-grid of $200 \times 200 \times 200$ was employed to perform the BZ integration.

## Data availability

The data that support the findings of this study are available from the corresponding author upon reasonable request.

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

## Acknowledgements

This work was supported by the National Key Research and Development Program of China (2022YFB4400200), National Natural Science Foundation of China (T2394474, T2394470, 92164206, 52121001, 12374011), the Science and Technology Major Project of Anhui Province Grant No. 202003a05020050, the New Cornerstone Science Foundation through the XPLORER PRIZE.

## Author contributions

W.S.Z. and S.J.X., led the project. S.J.X. performed sample growth and electrical and magnetic measurements, with assistance from W.S.Z., Y.H.J., K.L.W., and B.Q.D. Structural measurements were performed by S.J.X. and B.Q.D. Theoretical calculations were performed by Y.H.J. All authors contributed to the discussion of results. W.S.Z., Y.H.J., and S.J.X. wrote the manuscript. S.X. acknowledges a PhD scholarship by the China Scholarship Council (CSC).

## Competing interests

The authors declare no competing interests.
