## [Peer Review File · Nature Communications]

Reviewers' Comments:

Reviewer #2:

Remarks to the Author:

In their manuscript on the "Universal Scaling Law for chiral antiferromagnetism" the authors investigate the anomalous Hall response of Mn₃Pt both experimentally and theoretically. Understanding the electrical response of chiral antiferromagnets like Mn₃Pt is currently under active research. The challenge is to disentangle the various possible correlations with the complex magnetic texture and to understand which physical variables control the relative magnitude of the various possible channels for the anomalous Hall conductivity. Therefore, I believe that the findings are potentially very relevant to the field.

The line of argumentation presented by the authors can be seen as a sequence of 3 steps:

1. They experimentally observe a scaling law of the anomalous Hall resistance with the longitudinal resistance which can be described by a 2nd degree polynomial.
2. Referring to previous literature on electric transport in chiral antiferromagnets, the authors assign physical meaning to the coefficients of this polynomial.
3. The author's try to corroborate their findings using DFT calculations and by analyzing the thickness dependence of the observed effect.

Criticism could be made for each of these points. While the quality of the samples seems to be high and the obtained data seems to be valid, the rigor of the data analysis does not live up to the author's claim of discovering a "universal" scaling law. For the following reasons: it is very difficult to judge the quality of the supposed law solely based on Figure 3. No error estimates are made and the relevant data series for Mn₃Pt is tugged away in the corner where it is difficult to evaluate how closely the scaling law is actually traced by the experimental data. I would encourage the author's to improve this part of the presentation, since it is one of the core results of the paper.

Concerning the thickness dependence, it did not become very clear to me how the magnetic structure behaves as a function of the film thickness and how this correlates with the scalar spin chirality. For example, Figure 4 b) shows a schematic of how the magnetic structure might evolve. The problem is: in the 3-site model, the SSC is zero for both the purely ferromagnetic configuration and the purely antiferromagnetic configuration. It attains a maximum at intermediate canting angles. One would therefore expect a non-monotonous behavior of the SSC if the evolution of the magnetic texture would be as drastic as shown in Figure 4 b). I see this as another critical piece of the author's argument and would encourage them to lay out their argument more carefully with a special scrutiny on the evolution of the magnetic texture as a function of the film thickness.

And with regard to the DFT calculations it would be helpful to specify more transparently, how the magnetic texture was set up. I assume it was a purely antiferromagnetic state, which was then slowly canted towards 111-direction?

Overall, I believe that the results of the manuscript are potentially interesting and relevant to the field. But the quality of the presentation currently does not meet the standard of the journal and needs to be improved as it was laid out above. As of now, I would therefore not recommend the publication until those points have been addressed.

Reviewer #3:

Remarks to the Author:

In the manuscript entitled 'Universal scaling law for chiral antiferromagnet', by Shijie Xu et al., submitted to Nature Communications, the authors have studied the anomalous Hall effect (AHE) of

the chiral non-collinear antiferromagnet (AFM) Mn₃Pt films grown on MgO(001) substrate. The AHE, they observed, has two distinctive contributions; one is an intrinsic mechanism induced by a time-reversal symmetry breaking, and the second is a skew scattering induced by topological AHE due to a scalar spin chirality. To distinguish these two AHE contributions, they used a universal AHE scaling law, which was previously proposed to explain the AHE resistivity (ρ_{AHE}) in Mn₃Pt: $\rho_{\text{AHE}} = a_{\text{sk}} \cdot \rho_{\text{xx}} + b_{\text{in}} \cdot \rho_{\text{xx}}^2$ where ρ_{xx} is longitudinal resistivity and a_{sk} (b_{in}) is the skew scattering topological AHE (the intrinsic AHE). The main finding is that the both a_{sk} and b_{in} parameters change with the reversal film thickness d . They explained this behavior by considering the enhancement/reduction of the out-of-plane anisotropy due to spin canting, which is also supported by band calculations.

In my opinion, their finding and their conclusions are insufficient for publication in Nature Communications. Since they used the scaling law already proposed, the title of the manuscript "universal scaling law for chiral antiferromagnet" sounds very exaggerated.

There should be several steps and let me elaborate on these points below:

(1) The scaling is conducted by ρ_{AHE} and ρ_{xx} obtained around room temperature, which is unusual. In line number 76, the authors wrote "the new scaling law is universal for describing the AHE of chiral magnet". I could not find the "new scaling law" in the manuscript.

(2) The authors tried to elucidate the origin of the thickness dependence of AHE by using magnetic anisotropy. But this argument should rely on experimental evidence, ex magnetic properties. In Fig1, they presented an MH curve but only one thickness and only a c-axis magnetic field. Please present additional data of different thicknesses and different magnetic field axis (ex. In-plane). Otherwise, I can not believe their conclusion about the thickness dependence.

(3) The scaling law of AHE was proposed for zero temperature in theory. So the conclusions made only with room temperature measurements are not appropriate.

(4) In Fig.4, the authors present a_{sk} and b_{in} as a function of $1/d$. How did they get them? Please put an error bar here because their thickness dependencies are apparently small.

(5) Shape of the Ryz-H curves presented in Fig.2 is totally different from that of the M-H curve in Fig. 1. Please explain the reason.

(6) Did the author include spin-orbit coupling (SOC) in the band calculation? I guess that it might give a non-negligible effect because Pt has large SOC. At least, please mention with or without SOC in the calculation setup.

(7) I cannot find Fig.2e and Fig.2f.

1. A separate 'response to referees' letter that addresses the referees 2' in a point-by-point manner

Response:

Reviewer #2 (Remarks to the Author):

Criticism could be made for each of these points. While the quality of the samples seems to be high and the obtained data seems to be valid, the rigor of the data analysis does not live up to the author's claim of discovering a "universal" scaling law. For the following reasons:

it is very difficult to judge the quality of the supposed law solely based on Figure 3. No error estimates are made and the relevant data series for Mn3Pt is tugged away in the corner where it is difficult to evaluate how closely the scaling law is actually traced by the experimental data. I would encourage the author's to improve this part of the presentation, since it is one of the core results of the paper.

Response 1: Thank reviewer 2 very much for your suggestion. Your suggestion makes this article more scientific and rigorous. We added error bar at figure4 and revised figure3 according to your suggestion.

Figure 4.a-c The sheet resistivity ρ_{xx} , a_{sk} , the measured magnetizations M and b_{in} versus $1/d$

for Mn₃Pt. The inset in (a) shows the electronic scattering at the surface and the interface. The inset of (b) illustrates that there is larger SSC excitation in thicker films. The measurements were performed at 340 K. Solid lines refer to the linear fitting results.

Concerning the thickness dependence, it did not become very clear to me how the magnetic structure behaves as a function of the film thickness and how this correlates with the scalar spin chirality. For example, Figure 4 b) shows a schematic of how the magnetic structure might evolve. The problem is: in the 3-site model, the SSC is zero for both the purely ferromagnetic configuration and the purely antiferromagnetic configuration. It attains a maximum at intermediate canting angles. One would therefore expect a non-monotonous behavior of the SSC if the evolution of the magnetic texture would be as drastic as shown in Figure 4 b). I see this as another critical piece of the author's argument and would encourage them to lay out their argument more carefully with a special scrutiny on the evolution of the magnetic texture as a function of the film thickness.

Response 2: Thank reviewer 2 very much for your suggestion. We revised the figure 4 and show the measured magnetizations and SSC increased at larger film thickness. Based on our experiment data and DFT calculation, the SSC will increase at a smaller range and approach to the maximum value.

And with regard to the DFT calculations it would be helpful to specify more transparently, how the magnetic texture was set up. I assume it was a purely antiferromagnetic state, which was then slowly canted towards 111-direction?

Response 3: Thank reviewer 2 very much for your suggestion. We revised the figure 5 and shown how the magnetic texture was set up. The magnetic moment is initially set to a purely antiferromagnetic state, then we slightly tilt the magnetization of Mn atoms, the total net magnetization is set to the [111] direction of the Mn₃Pt lattice. And set up is shown in line V252 to 254 of the new manuscript.

Figure 5. (a-b) Band structure of R- Γ path for Mn3Pt, the color represent the spin polarization component on [111] direction. (c-d) Band structure near Fermi energy (upper panel) and Berry curvature (lower panel) in atomic units along the symmetry lines. The black line represents the purely antiferromagnetic state, the red line represents the state with net magnetization along [111] direction, whose z-component is $0.30 \mu_B$. (e) Calculated intrinsic contribution to the AHC versus Fermi energy with different net magnetizations. (f) Calculated intrinsic contribution to the AHC versus different total magnetizations at the Fermi level.

1. A separate 'response to referees' letter that addresses the referees 3' in a point-by-point manner

Response:

In my opinion, their finding and their conclusions are insufficient for publication in Nature Communications. Since they used the scaling law already proposed, the title of the manuscript "universal scaling law for chiral antiferromagnet" sounds very exaggerated.

There should be several steps and let me elaborate on these points below:

(1) The scaling is conducted by ρ_{AHE} and ρ_{xx} obtained around room temperature, which is unusual. In line number 76, the authors wrote "the new scaling law is universal for describing the AHE of chiral magnet". I could not find the "new scaling law" in the manuscript.

Response 1a: Thank reviewer 3 very much for your suggestion. Your suggestion makes this article more scientific and rigorous. For sure, ρ_{AHE} and ρ_{xx} obtained around zero temperature is better to conduct the scaling Law. However, it is hard to obtain the ρ_{AHE} below room temperature for Mn_3Pt due to larger saturation field at low temperature as shown at figure I. On the other hand, previous paper show that the scaling around room temperature and low temperature is the same and both of them is linear (1, linear scaling law at 5-320 K, *Physical review letters* 103.8 (2009): 087206. 2, linear scaling law at 5-300 K, *Physical Review Letters* 109.6 (2012): 066402). Finally, our experiment data also show the linear scaling law at 300-390 K, it should also be rigorous.

Figure I, Magnetic-field-dependent anomalous Hall resistance at 30 nm for chiral magnets Mn_3Pt , the measured temperature is 250 K.

Response 1b: Thank Referee 3 for his/her useful advice. The old scaling law ($\rho_{AH} = a\rho_{xx} + b_{in}\rho_{xx}^2$) can only describe the ferromagnetic materials, it ascribes the first term to the skew scattering by the impurity. However, our new scaling law ascribes the first term to the topological AHE due to an out-of-plane spin canting (topological anomalous Hall scaling, new physics). And we also first describe almost all the chiral antiferromagnetism (AFM scaling, new materials).

(2) The authors tried to elucidate the origin of the thickness dependence of AHE by using magnetic anisotropy. But this argument should rely on experimental evidence, ex magnetic properties. In Fig1, they presented an MH curve but only one thickness and only a c-axis magnetic field. Please present additional data of different thicknesses and different magnetic field axis (ex. In-plane). Otherwise, I can not believe their conclusion about the thickness dependence.

Response 2:

Thank Referee 3 for his/her useful advice. Your suggestion makes this article more rigorous. We have added experiment data of different thicknesses as shown in figure II and revised the Fig.4b.

Figure II,
Magnetic-field-dependent magnetization data at different thickness for chiral magnets Mn₃Pt, the measured temperature is 300 K.

(3) The scaling law of AHE was proposed for zero temperature in theory. So the conclusions made only with room temperature measurements are not appropriate.

Response 3: We explain it in Response 1a

(4) In Fig.4, the authors present a_{sk} and b_{in} as a function of $1/d$. How did they get them? Please put an error bar here because their thickness dependencies are apparently small.

Response 4: Thank Referee 3 for his/her useful advice. We get the a_{sk} and b_{in} by the linear scaling fitting ($\frac{\rho_{AH}}{\rho_{xx}} = a_{sk} + b_{in}\rho_{xx}$) at figure 3. We revise the figure 4 and add the error bar.

(5) Shape of the Ryz-H curves presented in Fig.2 is totally different from that of the M-H curve in Fig. 1. Please explain the reason.

Response 5: Thank Referee 3 for his/her useful advice. The M-H loop reflects the evolution of the net magnetic moment. The Ryz-H loop describes the evolution of the Berry curvature, which does not necessarily follow the net moment of the antiferromagnet. Also, in our article, we mention “ It is worth noting that the magnetic coercivity is much smaller than that of AHE coercivity, because the Zero magnetic moment indicates that the spin canting is zero. However, the AHE coercivity indicates that the topological AHE and the intrinsic AHE have opposite and equal contributions.”

(6) Did the author include spin-orbit coupling (SOC) in the band calculation? I guess that it might give a non-negligible effect because Pt has large SOC. At least, please mention with or without SOC in the calculation setup.

Response 6: Thank Referee 3 for his/her useful advice. Sure, we have considered the spin-orbit coupling (SOC) in the band calculation and the details can be found in the methods section.

(7) I cannot find Fig.2e and Fig.2f.

Response 7: Thank Referee 3 for his/her useful advice. we revised the Fig.2 and the article.

Reviewers' Comments:

Reviewer #3:

Remarks to the Author:

I have carefully read the revised manuscript and the author's response. I found some of their responses and manuscript revisions confusing. For instance, in response 2 to one of my comments regarding the lack of experimental evidence for thickness-dependent magnetic anisotropy, the authors presented a dataset. However, they did not provide a detailed explanation of how they reached their conclusion about the magnetic anisotropy, as depicted in Figure 4b. Additionally, in the main text (line 153), it is mentioned that the perpendicular magnetic anisotropy in the Mn₃Pt films changes with different thicknesses (Fig. 2b). However, I could not find this data in the manuscript, or at least, it was not apparent to me. These points are crucial for validating the new scaling law they claim, as they are connected to the scalar spin chirality that induces the topological anomalous Hall effect, which is the first term of their scaling law. Therefore, I kindly request a detailed explanation of the interpretation of the MH curve and how it relates to the evolution of scalar spin chirality as a function of thickness. Without this clarification, I believe that the work is not suitable for publication in Nature Communications.

Reviewer #4:

Remarks to the Author:

In my opinion, the authors have not sufficiently addressed the concerns raised in the previous round of review. This concerns mostly the following points:

1. The quality of the presentation of the data and the data analysis itself does not live up to the claim of a discovered "universal scaling law". It is still very difficult to judge whether or not the data actually supports their claim. Among other things this is made difficult also by the logarithmic scaling of the x-axis in Figure 3 and the barely visible data series.
2. At the same time, the scaling relation would not be an entirely new insight since for example reference [9] in the revised manuscript clearly describes skew scattering contributions to the AHE via the scalar spin chirality.

Based on this, I cannot recommend the publication in Nature Communications.

Reviewer #3 (Remarks to the Author):

I have carefully read the revised manuscript and the author's response. I found some of their responses and manuscript revisions confusing. For instance, in response 2 to one of my comments regarding the lack of experimental evidence for thickness-dependent magnetic anisotropy, the authors presented a dataset. However, they did not provide a detailed explanation of how they reached their conclusion about the magnetic anisotropy. Additionally, in the main text (line 153), it is mentioned that the perpendicular magnetic anisotropy in the Mn₃Pt films changes with different thicknesses (Fig. 2b). However, I could not find this data in the manuscript, or at least, it was not apparent to me. These points are crucial for validating the new scaling law they claim, as they are connected to the scalar spin chirality that induces the topological anomalous Hall effect, which is the first term of their scaling law. Therefore, I kindly request a detailed explanation of the interpretation of the MH curve and how it relates to the evolution of scalar spin chirality as a function of thickness. Without this clarification, I believe that the work is not suitable for publication in Nature Communications.

Point one: in response 2 to one of my comments regarding the lack of experimental evidence for thickness-dependent magnetic anisotropy

Point two: perpendicular magnetic anisotropy in the Mn₃Pt films changes with different thicknesses (Fig. 2b)

Point three: A detailed explanation of the interpretation of the MH curve and how it relates to the evolution of scalar spin chirality as a function of thickness

Thank Reviewer #3 for your suggestion. We have not only measured a large number of anomalous Hall data, but also supplemented magnetic measurements, even though the antiferromagnetic magnetic moment is very difficult to measure and very small.

For the point one about the antiferromagnetic magnetic anisotropy, few experimental groups have reported it. The magnetic anisotropy energy (due to spin canting effect) can be described as $K_A = |\int H_{\text{hard axis}} * dM - \int H_{\text{easy axis}} * dM|$. H is the magnetic field, M is the net moment. However, H. Chen et al. (Phys. Rev. Lett. 112, 017205) theoretically predicted the evolution of the magnetic structure about Mn₃Ir (which have the same structure of Mn₃Pt). The antiferromagnetic spin structure will tilt upward under positive saturation magnetic field and have the opposite result under negative magnetic field. When the large magnetic field was applied at Hard axis, the net moment go to zero $\int H_{\text{hard axis}} * dM = 0$, So the $K_A = \int H_{\text{easy axis}} * dM$. Therefore, the perpendicular magnetic anisotropy energy in the Mn₃Pt films with different thicknesses can described at figure 1. The magnetizations M induced by the net moment increases with the thickness, which means that the spin structure will tilt larger with the thickness. At a result, the scalar spin chirality will increase at larger thickness, causing the larger topological anomalous Hall effect.

Supplementary Figure 1: The net moment's perpendicular magnetic anisotropy energy in the Mn3Pt films with different thicknesses. Inset is the MH loop for chiral magnets Mn3Pt with different thickness at 300 K.

For the point two about the figure 2b, it the anomalous Hall effect change with thickness, not the perpendicular magnetic anisotropy. We have revised these contents. And the magnetic anisotropy was shown in the inset of Supplementary Figure 1.

For the point there, we revised the content in line 97-104 and line 231-234

line 97-104: The corresponding spin texture at different fields is shown in the illustration figure (Fig. 1c). Here, the SSC is characterized by the stacking direction of the atoms as viewed from [111] axis. When the external magnetic field is positive (greater than the saturation field), the antiferromagnetic spin structure will tilt upward. As a sequence, the SSC will be positive and have the opposite result under negative magnetic field. In addition, when the magnetic field is zero, the net moment and SSC will be non-zero due to magnetic hysteresis effect which will induce a topological AHE.

line 231-234: The magnetizations M induced by the net moment increases with the thickness, which means that the spin structure will tilt larger with the thickness. As a result, the scalar spin chirality will increase at larger thickness, causing the larger topological anomalous Hall effect

Reviewer #4 (Remarks to the Author):

In my opinion, the authors have not sufficiently addressed the concerns raised in the previous round of review. This concerns mostly the following points:

1. The quality of the presentation of the data and the data analysis itself does not live up to the claim of a discovered “universal scaling law”. It is still very difficult to judge whether or nor the data actually supports their claim. Among other things this is made difficult also by the logarithmic scaling of the x-axis in Figure 3 and the barely visible data series.
2. At the same time, the scaling relation would not be an entirely new insight since for example reference [9] in the revised manuscript clearly describes skew scattering contributions to the AHE via the scalar spin chirality.

Point 1: Thank Reviewer #4 for your suggestion. We have not only measured a large number of anomalous Hall data to confirm the universal scaling law, but also analyzed almost all the antiferromagnetic materials which have the anomalous Hall effect. In addition, our theories and experiments show almost identical results, explaining in detail the relationship between the anomalous Hall effect and the antiferromagnetic spin structure. This work clarifies the origin of the anomalous Hall effect in antiferromagnetism.

We also have added linear scale relationships to the supplementary material

Supplementary Note 2: Linear scaling law in the MnGe films.

Supplementary Figure 2 | Universal scaling law for chiral magnets MnGe [2]. The various chiral AFM was plotted by $\frac{\rho_{AH}}{\rho_{xx}} = a_{sk} + b_{in}\rho_{xx}$. The slope and intercept represent the intrinsic and skew scattering anomalous Hall scaling factors, respectively.

The MnGe shows the linear (ρ_{xy} / ρ_{xx}) scaling relation for the anomalous Hall resistivity. In addition, the slope component is Zero which means the skew-scattering in MnGe is dominated and the intrinsic contribution should be negligible. Both the 160 nm and 80 nm MnGe show the same linear scaling Law. One possibility for this unconventional skew scattering is the recently proposed “spin-chirality skew-scattering” mechanism [3], which caused by scalar spin chirality (SSC) or out-plane spin canting effect.

Supplementary Note 3: Linear scaling law in the Mn₃Sn, Mn₃Ir, Mn₃Ge films.

Supplementary Figure 3 | Universal scaling law for chiral magnets Mn₃Sn [5], Mn₃Ir [4], Mn₃Ge [5]. The various chiral AFM was plotted by $\frac{\rho_{AH}}{\rho_{xx}} = a_{sk} + b_{in}\rho_{xx}$.

The Mn₃Sn, Mn₃Ir and Mn₃Ge show the linear (ρ_{xy} / ρ_{xx}) scaling relation. For the Mn₃Sn and Mn₃Ge, the intercept component is Zero which means the intrinsic mechanism is dominated and the skew scattering should be negligible. Because the Mn₃Sn and Mn₃Ge only have the in-plane spin canting effect (SSC=0) and the anomalous Hall effect come from intrinsic non-zero berry phase. For the Mn₃Ir film, both the intrinsic mechanism and skew scattering have non-negligible contribution.

Supplementary Note 4: The measured Linear scaling law in the Mn₃Pt films.

Supplementary Figure 4 | Universal scaling law for chiral magnets Mn₃Pt with the different

thickness d . The various chiral AFM was plotted by $\frac{\rho_{AH}}{\rho_{xx}} = a_{sk} + b_{in}\rho_{xx}$.

Both the Mn₃Ir and Mn₃Pt have the same spin structure which have the large intrinsic anomalous Hall effect [1,6]. In addition, the corresponding spin texture at different fields is shown in the illustration figure (Fig. 1c, main text). The antiferromagnetic spin structure will tilt upward or downward under saturated magnetic field. As a sequence, the SSC will be non-zero and induce a topological skew scattering AHE at the same time.

Point 2: Thank Reviewer #4 for your suggestion. Reference [9] only think that the skew scattering dominates the AHE for MnGa. However, this work can't explain all the AFM materials. Our works have not only proposed the universal scaling law, which can get the intrinsic contribution and skew scattering contribution, but also analyzed the relationship between AHE and AFM spin texture. Both the scalar spin chirality and vector spin chirality should have contribution for AHE in AFM materials. In addition, our ab initio calculation and experiments show almost identical results.

Reviewers' Comments:

Reviewer #3:

Remarks to the Author:

The authors addressed all my comments, and I think that the manuscript is now ready for publication in my opinion.